# Recent Advances in RNA Therapy and Its Carriers to Treat the Single-Gene Neurological Disorders

**DOI:** 10.3390/biomedicines10010158

**Published:** 2022-01-12

**Authors:** Ming-Jen Lee, Inyoul Lee, Kai Wang

**Affiliations:** 1Department of Neurology, National Taiwan University Hospital, Taipei 10012, Taiwan; mjlee@ntu.edu.tw; 2Department of Medical Genetics, National Taiwan University Hospital, Taipei 10012, Taiwan; 3Institute for Systems Biology, Seattle, WA 98109, USA; il@systemsbiology.org

**Keywords:** single-gene disorder, neurological disease, RNA therapy, mRNA, RNA interference, carriers

## Abstract

The development of new sequencing technologies in the post-genomic era has accelerated the identification of causative mutations of several single gene disorders. Advances in cell and animal models provide insights into the underlining pathogenesis, which facilitates the development and maturation of new treatment strategies. The progress in biochemistry and molecular biology has established a new class of therapeutics—the short RNAs and expressible long RNAs. The sequences of therapeutic RNAs can be optimized to enhance their stability and translatability with reduced immunogenicity. The chemically-modified RNAs can also increase their stability during intracellular trafficking. In addition, the development of safe and high efficiency carriers that preserves the integrity of therapeutic RNA molecules also accelerates the transition of RNA therapeutics into the clinic. For example, for diseases that are caused by genetic defects in a specific protein, an effective approach termed “protein replacement therapy” can provide treatment through the delivery of modified translatable mRNAs. Short interference RNAs can also be used to treat diseases caused by gain of function mutations or restore the splicing aberration defects. Here we review the applications of newly developed RNA-based therapeutics and its delivery and discuss the clinical evidence supporting the potential of RNA-based therapy in single-gene neurological disorders.

## 1. Introduction

Over the past 30 years, knowledge of neurogenetics has evolved considerably. Advances in sequencing technology and molecular medicine have led to the identification of many genes that are involved in neurological diseases. Functional studies have revealed new insights in biochemistry and biological processes that are associated with disease pathophysiology. Based on these molecular insights and the technology breakthrough in therapeutic developments, the treatments for previously incurable diseases were proven to be feasible and will accelerate in the coming decade. Given the progress in genetic diagnosis and understanding of the pathophysiology of neurological disorders, especially single gene disorders, we aim to summarize the recent development of targeted ribonucleic acid (RNA)-based therapy and its deliveries in this review.

The human genome contains three billion deoxyribonucleotides and is organized into 23 pairs of chromosomes [1,2]. Genes are deoxyribonucleic acid (DNA) sequences that are transcribed into RNAs and subsequently translated into proteins. This is the central dogma of biological systems—from prokaryotic to eukaryotic cells. Each cell contains two copies of the genome, one from each parent. Rare genetic diseases often follow the Mendelian transmission patterns and are usually the direct result of a single gene mutation. In autosomal dominant inheritance, only one mutated allele in the causative gene is needed to cause the disease. The autosomal dominant neurological diseases include Huntington’s disease, familial amyloid polyneuropathy, neurofibromatosis, myotonic dystrophy, etc. Autosomal recessive diseases need two mutated alleles which can come from the asymptomatic or mildly symptomatic parents (e.g., Wilson’s disease, Friedrich ataxia, Tay-Sachs disease, etc.). In addition to autosomal transmission, a few diseases are linked to mutations in genes on the sex chromosomes. X-linked recessive diseases occur more commonly in male family members since males have only one X chromosome. Female members are usually mildly symptomatic or asymptomatic carriers. There is no male–male transmission in the X-linked diseases (e.g., Charcot-Marie-Tooth disease X, Duchene muscular dystrophy, etc.). DNA replication errors at the developmental stage can cause spontaneous mutations or mosaicism in somatic or germline cells. The reduced penetrance, differential expressivity, pleiotropy, or de novo mutation can cause difficulties in differentiating transmission patterns.

A single nucleotide change can result in a silent, missense, or nonsense mutation. A silent mutation does not change the amino acid residue, but a missense mutation does. Depending on the impact of mutated amino acid residue, a missense mutation can cause haploinsufficiency, a gain of function, or a dominant negative effect. A nonsense mutation can lead to a stop codon which results in premature termination of protein translation. A frameshift mutation that is caused by insertion, deletion, or rearrangement of the DNA sequences can also disrupt the entire reading frame of a gene as well as introduce a premature stop codon, which results in a severe dysfunctional protein. Haploinsufficiency or loss of function that is caused by the truncated protein is frequently encountered in tumor suppressor proteins (e.g., p53 and PTEN) [3,4] or enzyme deficiencies (e.g., Pompe disease, lysosomal storage disease) [5,6]. Another category of mutations is microsatellite repeat expansion (MRE) which is responsible for several neurological disorders. Toxic gain of function is the main molecular pathology that is associated with MRE, which can take place at either coding or noncoding regions and its size can further expand in successive generations. The size of MRE results in progressively toxic RNA or protein products and correlates with disease severity and age of onset [7]. Mutations in the noncoding regions can be detrimental to cellular function if they are involved in epigenetic control such as preventing the binding of a transcription factor(s) resulting in aberrant expression. On some occasions, enhancers can be located far from their target genes and form long chromatin loops to modulate the transcription machinery [8]. Mutations in the enhancer regions may affect the expression of their target gene(s).

The molecular mechanisms for the diseases include loss of function, haploinsufficiency, and toxic gain of function. Understanding the molecular pathology would help in designing treatment strategies. Recently, the advances in precision medicine and targeted therapy have been nourished by the progress in sequencing technologies and biostatistical analysis. The alteration of a disease-associated gene in either the concentration or its sequence attracted attention in recent years. The manipulation at the transcription level becomes feasible with the introduction of the eukaryotic expression system and RNA interference technology. In addition to the direct manipulation on the level of a specific transcript, the idea to modify the splicing process to preserve the partial function of the very protein which is pivotal to physiological needs has been introduced. Furthermore, the newly designed nanocarriers to facilitate the delivery and release of the nucleic acids without causing immune responses has also made progress in recent years. Thus, RNA therapy to single-gene neurological disorders in routine clinical practice is on the horizon.

## 2. RNA-Based Therapy

The use of nucleic acid therapy to treat monogenetic diseases is on the horizon of genomic medicine. Based on the preclinical studies in cellular and animal models, the emergence of therapeutic messenger RNAs (mRNAs) and short RNAs was proven to be a new class of medicine in a spectrum of diseases [9], such as cardiovascular diseases [10], oncology/immunology [11,12], and infectious diseases [9].

### 2.1. Therapeutic mRNA for Protein Replacement

For diseases that are caused by haploinsufficiency or loss of function, replacing the defective protein is a straightforward treatment strategy. Proteins that are synthesized by bacteria or cultured cells, or synthetic peptides are commonly used to supply or replace the defective proteins. Even though the use of recombinant protein extended the arsenal of treating some single gene neurological diseases, there are many drawbacks including the high cost, potential immunogenicity, and only suitable for some disease entities [13]. RNA-based therapeutics offers an alternative. For example, the wild-type gene can be introduced into cells to replace the defective disease-causing gene with an engineered virus; however, some hurdles of using viral carriers remain difficult to overcome. Strong humoral immune responses that are triggered by viral vectors may prevent repeat administration. The levels of genes that are introduced to cells, ranging from an ineffective low copy to a toxic high level can cause the variability of treatment efficiency. In addition, the choice of viral vector as well as the copy number of enclosed mRNA need to be optimized before any clinical development.

An alternative to use a viral vector for defective gene replacement is to introduce chemically-modified mRNA directly into the cells. This approach has been adapted in oncology as well as cardiovascular and infectious diseases [9,14]. The mRNA-based therapeutics have also been tested in a few monogenetic neurological disorders [15,16]. mRNA-based therapy transfers genetic information and transiently expresses the functional protein in the cytoplasm without affecting the patient’s genome. Although it can avoid possible inherent genotoxicity of viral vector-based gene therapy, the short, transient expression of mRNA therapeutics may require repeated administration to treat or manage chronic neurological diseases which may be cost prohibitory.

Structurally, the mRNA is composed of a 5′ cap followed by a 5′ untranslated region (UTR), a protein coding region, a 3′ UTR, and a poly (A) tail. The 5′ cap, UTR regions, as well as the poly (A) tail are important for stabilizing the mRNA [17,18]. Modifying sequences in these regions may affect the stability and translatability of the mRNA. For example, the introduction of microRNA binding sites into the 3′ UTR can modulate the stability of mRNA in cells [19]. In addition, the efficiency of translation can be enhanced with the suitable Kozak consensus sequence ((GCC)GCCRCCATGG) and the improvement of the secondary structure [20]. The modified nucleotides such as N^1^-methylpseudouridine or 5-methoxyuridine can also be used to increase the protein yield. Moreover, the modified mRNA can reduce the activation of immune RNA sensors such as the retinoic acid-inducible gene I and the Toll-like receptors [21]. The inactivation or reduction of immune responses toward the therapeutic RNAs is crucial for allowing repetitive administrations to treat chronic diseases.

### 2.2. Single-Stranded Oligonucleotide as a Therapeutic Agent

In addition to full-length mRNA for protein replacement therapy, short single stranded oligonucleotides or chimeric RNA-DNA have been explored as a therapeutic modality [22]. Single-stranded oligonucleotides are synthetic, short RNA, or DNA molecules. Modified nucleotides can be incorporated to enhance their functionality and stability. Their functions are based on either the complementarity with the target; for example, antisense oligonucleotides (ASOs) and immune stimulatory oligonucleotides (ISOs), or the tertiary structure such as aptamers [23]. The activities of ASO therapeutics are directed by their ability to form Watson and Crick base-pairing with their target sequences [24], whereas ISOs are recognized and bound by proteins or antibodies that detect specific sequence motifs [25]. The first ASO therapy, Formivirsen, consists of a 21 nucleotide long DNA molecule with a phosphorothioate (PS) backbone [26] and was approved in 1998 to treat cytomegalovirus. The second ASO drug, Mipomersen was for homozygous familial hypercholesterolemia (HoFH), which forms a double-strand complex with apolipoprotein B (APOB) transcript and causes its degradation through ribonuclease RNase H [27]. The new generation of ASOs employs new chemical modifications and designs that lead to safer (reduced immune-related inflammation responses), more stable (increased nuclease resistance), and more potent (enhanced RNA binding affinity) compounds with improved pharmacokinetics and pharmacodynamics properties. The maturation of the field has led to the increase of approved single stranded oligonucleotide therapeutics in recent years. There are six new ASO therapies and one ISO therapy that have obtained approval from either the Food and Drug Administration (FDA) or European Medicine Agency (EMA) in the past four years [22]. These therapies are aimed at diseases including neurological and metabolic/cardiovascular diseases, muscular dystrophy, and vaccines against viruses such as hepatitis B.

Based on the mode of action, there are three major types, i.e., gapmers, steric blocking, and CpGs. Gapmers are 16–20 nucleotides (nts) oligomers with a central stretch of 8–10 DNA nts flanked by a stretch of 4–5 nts on each side with modifications on the sugar rings. The sugar modifications can increase binding affinity, avoid or decrease immune activation, and increase resistance to nuclease. The PS bonds in the sugar ring of gapmers can enhance the stability and increase binding to plasma proteins [28]. Gapmers bind in a sequence-specific manner to their target RNA and form a double-stranded RNA-DNA hybrid, which can be recognized and initiate RNase H-mediated cleavage of the target RNA to reduce the transcript and its encoded protein level. Through a natural endocytic process, gapmers have been taken up by cells in the brain, eye, liver, kidney, adrenal glands, and lungs based on the literature [22]. Once in the cells, they are released into the cytoplasm from endosomes; however, it remains unclear how the gapmers enter into the nucleus [25]. Accordingly, gapmers can be employed to reduce the level of any protein or protein isoforms, as well as toxic proteins that are caused by dominant negative or gain-of-function mutations [29]. Up to date, three gapmers have been approved for clinical use, Mipomersen, Inotersen, and Volanesorsen. In addition to the single stranded oligonucleotides, double stranded DNA-RNA heteroduplex oligonucleotides (HDO) [30,31], and overhanging heteroduplex oligonucleotides (ODO) [32] were recently reported to have more potent gene silencing activities.

Steric-blocking ASOs are single-stranded oligonucleotides which are 15~30 nts in length. Both the sugar ring and the backbone of the nucleotides can be chemically modified. Similar to gapmer ASOs, the chemical modifications are intended to increase the nuclease resistance and binding affinity, as well as to reduce immune responses. Steric-blocking ASOs bind to the target RNA via sequence-specific base-pairing. The complementary pairing hinders the binding of trans-acting factors such as small nuclear RNAs, microRNAs, long non-coding RNAs, or RNA binding proteins, to their cognate sequences. The pairing between the steric-blocking ASOs and the target RNA also prevents the formation of proper RNA secondary structure [24,29]. The first steric-blocking ASO targeted the 3′ splice site of the herpes simplex virus type 1 pre-mRNA to prevent its normal splicing and function in the mid-1980s [33]. After that, the utilizing of steric-blocking ASOs to correct mutation-driven splicing defects and to modulate mRNA stability with subsequently amending protein translation increased [34]. Similar to gapmer ASOs, steric-blocking ASOs can be taken up by cells in the brain, eye, liver, kidney, adrenal glands, and muscles [22]. A few steric-blocking ASOs, including Nusinersen, Eteplirsen, Golodirsen, Viltolarsen, and Milasen, have been developed and used clinically [35]. Several steric-blocking ASOs-based clinical applications have emerged recently, after the development of chemically-modified nucleotides and the introduction of tissue targeting approaches [35]. ASO-mediated restoration of the aberrant splicing mutations has been introduced for several neurological diseases including spinal muscular atrophy and Duchene muscular dystrophy (details in the following sections).

A different application for steric-blocking ASO is exon-skipping to eliminate the exons with a missense mutation in the final mature mRNA. This strategy was first proposed for Emery-Dreifuss muscular dystrophy, a disease that is caused by mutations in the lamin A/C gene (LMNA) [36]. There is an alpha-helical central rod domain that is composed of multiple heptad repeats. Skipping the exon 3 or 5 will remove six repeats, yet still preserve the alpha-helix structure. The proof-of-concept of using a steric-blocking ASO to skip mutation containing exon 5 (Lamin A/C–Δ5) was reported to rescue laminopathy [36]. The protein without exon 5 functions as effectively as the wild-type Lamin A/C. Similar to LMNA, one of the responsible genes for Usher syndrome type II (USH2), the *USH2A* gene encoding usherin protein contains multiple repetitive domains. A deletion mutation in exon 13 of the USH2A gene occurs in 30% of USH2 patients. Pendse et al. hypothesized that the removal of one or more of the repetitive domains that is encoded by mutant exon(s) may restore the normal function of the USH2A protein [37]. An antisense oligonucleotide-based exon-skipping therapeutic, QRX-421 from ProQR therapeutics, for the USH2 that is caused by mutations of the USH2A gene, was FDA-approved in 2017.

Recently, a novel approach named targeted augmentation of nuclear gene output (TANGO) was introduced to use antisense oligonucleotides to modulate non-productive splicing events and enhance the level of functional transcript/protein [38]. Lim et al. demonstrated that using the TANGO platform can increase the endogenous full-length functional transcript levels by preventing naturally occurring non-productive alternative splicing events [38]. Increasing the level of ASO-mediated full-length functional mRNA correlates with the abundance of targeted nonsense-mediated decay-inducing events [38]. Thus, the TANGO platform is an alternative strategy to increase the concentration of functional mRNA and protein to reverse the pathology of autosomal dominant haploinsufficiency diseases. The TANGO strategy has been applied in the treatment of Dravet syndrome (DS), a rare autosomal dominant drug-resistant epilepsy that is caused by mutations in the SCN1A (Voltage-Gated Sodium Channel Alpha Subunit 1) gene. In a preclinical mouse model, the ASO, STK-001, prevented the inclusion of a mutated Scn1a gene exon leading to the increase of functional Scn1a mRNA concentration and restoring the pathology [39].

In summary, therapies utilizing single-stranded oligonucleotides represent a platform of precision medicine to treat genetic diseases that were once deemed untreatable. The innovation on the modification chemistries in the ASO drugs makes them stable, safe, and effective to be used in patients. Together with targeted delivery, ASO-based therapeutics are expanding the breadth in precision medicine and paving the way for clinical applications in previously untreatable genetic diseases. Effectively delivering the therapeutic agents to the target site is another important issue. For example, there are blood-brain and blood-nerve barriers that separate the nervous tissues from the systemic circulation. The next section will discuss the newly developed drug carriers which overcome these obstacles and can release the drug at or nearby the disease site.

## 3. Blood Brain Barrier and Molecular Carriers

### 3.1. Blood Brain Barrier

Anatomical and metabolic barriers between the brain parenchyma and the peripheral blood circulation maintain the proper activities of the central nerve system (CNS). These barriers include cerebrospinal fluid (CSF), choroid plexus, and the blood brain barrier (BBB). The BBB is a protective anatomic structure that prevents the brain from direct contact with the blood [40], an important component of the neurovascular unit in the CNS. It restricts substance exchanges between peripheral blood and the brain cells. The anatomical components of the BBB include endothelial cells, end-feet links of astrocytes, basal lamina, tight junctions, and pericytes [41,42,43]. The tight junctions (junctional adhesion molecules), as well as adherens, gap junctions, pericyte endothelial junctions, astrocyte junction, and the basement membrane all contribute to the separation between peripheral blood and the brain cells [44,45].

To move molecules across BBB is usually through either a carrier-mediated or a receptor-mediated transport (RMT) system. An array of RMT transporters, receptors, and channels on endothelial cells and pericytes are involved in the process [46]. Thus, the BBB plays a major role to govern the transport of neurological factors from the blood, allowing only beneficial and nourishing substances to pass and protect the brain cells from toxic substances [47]. Nutrients need to cross the capillary endothelial plasma membrane via the efflux systems, active transporters, and ectoenzymes to enter the brain [48]. To cross the BBB, there are a few general determinants, such as molecular weight (<400 Da), morphology (spherical), size (nanometer range), ionization (pH value), and lipophilic properties, etc. [49]. In addition, the associated peripheral factors such as enzymatic stability, the binding affinity of plasma proteins, the volume of distribution, intracranial pressure, clearance rate, and the influence of cytochrome P450 on the rate of oxidative metabolism, may contribute to the efficiency of BBB transportation [50,51]. The active transport process that is associated with the BBB is highly correlated with neuronal activities; therefore, it is an important topic during drug development for neurological disorders, such as neurodegenerative diseases.

### 3.2. Brain Drug Delivery Strategies

To adopt a new therapeutic entity into the clinic, several key data including toxicity profile, ADME (absorption, distribution, metabolism, and excretion) characteristics, and dosage need to be generated. For CNS drugs, the BBB is an additional factor that needs to be addressed; therefore, the delivery system is especially important for therapeutics that are used in neurological diseases. Though drugs can be delivered to the brain by local injection, via a catheter, or direct administration following an invasive surgery [52,53], these practices are not feasible in most clinical settings. Thus, a less invasive route with a steady therapeutic effect is required for CNS drugs. An alternative route is an intranasal administration which allows the drugs to reach the brain through the olfactory system and bypass the BBB. Nonetheless, intranasal administration depends solely on the absorption of the drug through the nasal mucosa, which is not an ideal approach due to inconsistent drug concentration at the target site [53,54]. Therefore, delivery through a systemic administration remains the most common approach for CNS drugs. There are a few criteria for a suitable CNS drug delivery system: (1) to target the brain cells with the capability to pass through the BBB; (2) to release the drug in the brain at a controlled pace; (3) to use a system that exhibits a desired pharmacologic activity; and (4) to employ safe and biocompatible materials, such as lipids or polymers. For example, the use of nontoxic and permeable nanoparticles to encapsulate active drugs can cross the BBB and deliver effective therapeutic dosages to the brain [55]. In addition, approaches to increase the permeability of the BBB such as injection of hyperosmolar mannitol to cause a reversible BBB disruption, or applying ultrasound as a temporary physical disruption of the BBB has been reported [56,57]. However, these approaches may introduce the influx of neurotoxin and cause significant damages to the brain [58]. Recently, lipid-based nanocarriers, extracellular vesicles, and adeno-associated viral carriers are the focus of BBB-penetrating drug delivery systems.

#### 3.2.1. Lipid-Based Nanocarriers

The growing field of nanoscale delivery systems for therapeutics is the main focus of basic research [59]. Nanocarriers for drug delivery may provide a more favorable half-life, a better-controlled release, and a precise localization [60,61,62]. Based on the material that is used, there are three categories of nanocarriers: organic-based, inorganic-based, or a combination of both. The organic nanocarriers include polymeric frameworks and lipid-based systems. Drug conjugates and micelles belong to the polymeric framework, while liposomes, nanoemulsions, and dendrimers are lipid-based systems. The metallic structures, silica nanoparticles, and quantum dots consist of inorganic nanocarriers [63]. Among the nanocarriers, the lipid-based nanoparticle is the leader in terms of higher levels of biocompatibility and flexibility. The potential advantages and short-comings of the lipid-based nanocarriers have been listed in Table 1.

The lipid nanoparticles (e.g., solid lipid nanoparticles, SLC) are the pioneer of this rapidly evolving field. There is a long list of attributes and advantages of lipid-based drug delivery systems such as controlled and targeted drug release, pharmaceutical stability, capability for incorporation of both lipophilic and hydrophilic drugs, biodegradability and biocompatibility, low-risk profile, better drug absorption, and lowering the therapeutic dose, etc. The structural modifications of lipid-based nanoparticles enable the delivery carrier to safely and effectively pass through the BBB [64]. Vehicle-based, emulsion-based, or particulate systems are the main lipid nanoparticle drug delivery tools in clinical use [63]. Lipid nanoparticles can satisfy different requirements based on the disease condition, route of administration, product stability, toxicity, and bioavailability. In addition, lipid nanoparticles also have the potential to achieve the goal of controlled and site-specific drug delivery. According to the chemical composition of lipids, lipid nanoparticles can be divided into three categories: homo-lipids, hetero-lipids, and complex lipids [65]. Formulations of lipid carrier-based drug delivery systems include superficial fluid-based methods, adsorption onto a solid carrier, spray drying, melting granulation, and congealing.

One of the key elements for RNA-based therapy is a carrier that can protect the RNA from nuclease degradation and penetrate the BBB with adequate bioavailability. Lipid nanoparticles are usually composed of phospholipids, sterols, and ionizable lipids. Adding polyethylene glycol (PEG)—conjugated lipids can reduce the capture by phagocytes after systemic administration. In clinical trials, the liver is the main organ for the distribution of lipid nanoparticles. By interacting with the low-density lipoprotein (LDL) receptor, apolipoprotein E, and other opsonins, the nanoparticles are internalized into the hepatocytes [66]. With chemical modifications, lipid nanoparticles can achieve better distribution to the target organ and escape entrapment with subsequent degradation in the endo/lysosomal compartment [67]. With endosomal escape, mRNA cargo can be released into the cytoplasm [68]. To replace a fully functional protein, it is important to avoid lipid accumulation and to improve the stability of the mRNA that is encapsulated in the lipid nanoparticles. To date, the nanocarriers that are used for the delivery into CNS are composed of synthetic or natural polymers such as PEG, PLA, PLGA, chitosan, etc. [69,70,71,72]. The surface, as well as the polymeric structure modifications of the particles facilitate them to enter into the brain and to unload the encapsulated RNAs in a controlled manner. Other potential issues that are associated with nanoparticle-based therapeutics delivery is the toxic metabolic by-products of the nanostructure, for example, polybutylcyanoacrylate (PBCA)-based nanoparticles [73]. In addition, the size of the synthetic nanoparticles can be as large as the cells, which may interfere with the normal function of the brain [74,75].

Solid lipid nanoparticles (SLN) have recently gained more attention as a drug delivery system [76]. SLNs are lipid-based biocompatible nanocarriers that are composed of lipid or modified-lipid nanostructures (10–1000 nm in size). There is a solid hydrophobic core that allows the dispersion of either hydrophilic or lipophilic drugs [77,78]. Another beneficial character is its capability in crossing the reticuloendothelial system (RES) of the BBB [78,79]. Its solid lipid composition plays a role as a safeguard to avoid the biochemical degradation of active drugs [80]. The SLNs are formed with a physiological solid lipid emulsion without organic solvents and they have better biocompatibility with reduced systemic toxicity compared to polymeric nanoparticles [81]. To obtain a sustained release feature of the drug, the solid lipids, the modified drugs, and the additive ingredients can be mixed in a particular ratio to provide a specific physicochemical state for a long diffusion process and the controlled release of the encapsulated drugs [82,83,84]. The advantages of SLNs include maximum drug bioavailability after administration, good control release kinetics, better tissue targeting, reduced immune responses, the ability to penetrate the BBB with little neurotoxicity, the ability to deliver both small molecules and biomolecules, adequate drug loading, good patient compliance, cost effective, and scalability for mass production [49].

#### 3.2.2. Extracellular Vesicles

Extracellular vesicles (EV) were discovered 30 years ago [85]. Recently, the interest in these small circulating vesicles has grown exponentially. EVs are small lipid vesicles and are involved in cellular communication, disposal of cellular waste, and transferring nucleic acids, and proteins. Based on the biogenesis, EVs in eukaryotes can be divided into three main categories, exosomes (30–100 nm), microvesicles (100–1000 nm), and apoptotic bodies (1–5 μm) [86,87]. Exosomes are formed through the endosomal system [88]. Microvesicles are established by outward budding followed by fission of the plasma membrane and then are released to the extracellular space [88]. The size of apoptotic bodies is generally larger than exosomes and microvesicles, which are formed during the process of programmed cell death and may contain fragmented nucleic acids, organelle debris, and nuclear components. Macrophages remove the apoptosis bodies by phagocytosis [89]. EVs are reported to be involved in a wide range of physiological and pathophysiological processes in neurological diseases. Their properties make them an ideal source for biomarkers and prospective drug delivery carriers.

The cargos that are carried by EVs include metabolites, DNA, proteins, and RNA [90,91,92,93]. Different types of RNAs including protein-coding mRNAs, noncoding lincRNAs, and small RNAs have been found in EVs. Some of the RNAs retain their functionality and cause alterations of the physiological state of the recipient cells [94]. In both in vitro [95] and in vivo [91] models, the exchange of genetic materials, RNA, and proteins by EVs has been confirmed. There are a few advantages to EV-based drug delivery. Firstly, studies showed that EVs are capable of withstanding many freeze-thaw cycles, rendering them easier to store and transport [96,97,98]. Secondly, evidence showed that EVs can increase the stability, bioavailability, and activity of the encapsulated drug. For example, curcumin that is encapsulated in exosomes is more stable and more concentrated in circulation [99]. Thirdly, EVs have advantages with reduced immunogenicity, inherent tissue specificity, and easy to scale production, as compared to well-established nanoscale delivery vehicles, such as nanoparticles or liposomes [100,101,102,103]. Small pharmacological agents or biological cargos such as RNA or proteins may be actively or passively packed into EVs [104]. Finally, the capability of EVs to cross the BBB make them a good candidate for delivering drugs to treat neurological diseases. A recent study showed that the use of EVs from brain endothelium successfully delivered functional anti-cancer therapy in a zebrafish brain cancer model [105]. The uptake of nanoparticles or EVs is mainly through endocytosis [106,107,108].

#### 3.2.3. Viral Carriers

The viral genome can be modified as a carrier to insert genetic materials into cells. As an example, using the gamma-retroviral vector to transduce hematopoietic stem cells to treat adenosine deaminase deficiency-induced severe combined immunodeficiency was approved by the EMA in 2016 [109]. The FDA approved the use of a recombinant lentivirus to modify the autologous T cells to treat chronic lymphoid leukemia in 2018 [110]. Furthermore, a modified herpes virus to deliver granulocyte-macrophage colony stimulating factor has been approved for the treatment of advanced melanoma [111]. The safety of an adeno-associated virus (AAV) carrying Retinoid Isomerohydrolase RPE65 (RPE65) cDNA for treating Leber’s congenital amaurosis (LCA) has been investigated and the results showed that the treatment does not pose any major adverse effect [112]. Nathwani et al. described that peripheral-vein infusion of a modified AAV carrying a codon-optimized human factor IX (F9) can induce the F9 transgene expression to improve the bleeding phenotype in patients with hemophilia B [113].

## 4. RNA Therapies to Treat Single-Gene Neurological Disorders

In this section, a few RNA therapeutic agents targeting single-gene neurological disorders will be introduced. These neurological disorders involve the CNS and PNS as well as multi-systems, although they are caused by mutations from single genes. Given the progress in RNA therapy, a few RNA drugs have been approved by FDA in the US (Table 2). Despite these agents, more are on the clinical trials which will also be introduced in the following sections.

### 4.1. Familial Amyloid Polyneuropathy

Hereditary amyloidosis is a rare, life-threatening, autosomal dominant, multi-organ disease that is characterized by amyloid deposits in multiple organs. Mutations in several genes are involved in hereditary amyloidosis and the most common one is the mutations in the transthyretin (TTR) gene [114]. The primary source of circulating transthyretin protein is from the liver. Nonetheless, amyloid deposits can be found in the heart, kidney, liver, gastrointestinal tracts, blood vessels, and peripheral nerves in hereditary transthyretin amyloidosis [114,115]. The symptoms from familial amyloid polyneuropathy (FAP) and cardiomyopathy (FAC) occur more commonly in patients with transthyretin amyloidosis [114,116]. Small fiber neuropathy involving the autonomic nervous system leads to orthostatic hypotension, chronic constipation and/or diarrhea, impotence, and bladder dysfunction, which are commonly found in patients with FAP [117,118,119]. As the disease progresses, FAP patients develop marked axonal degeneration, manifested by significant muscle wasting and weakness on the intrinsic hand and foot muscles, gait disturbances, and sensory deficits on distal limbs. Polyneuropathy leads to profound sensorimotor disturbances with the deterioration of daily living function and ambulation [117]. The main culprit for sudden death in hereditary transthyretin amyloidosis is cardiac involvement that is manifested by heart failure, orthostatic hypotension, severe conduction defects, and arrhythmia [120,121].

In 2012, Su et al. described that mice without TTR expression are viable [122]. Thus, suppressing the expression of transthyretin may be an effective way to treat hereditary amyloidosis. RNA interference (RNAi) to silence the TTR transcript is an approach to fulfill the need to suppress the transthyretin. Patisiran is the first FDA-approved RNAi therapeutic agent to reduce the production of both wildtype and mutated transthyretin by binding to the 3′UTR region of TTR mRNA [123]. Both the phase 3 trial, and the long-term safety and efficacy evaluation have shown that Patisiran is an effective treatment for FAP [124,125]. Another drug, Inotersen is a 2′-o-(2-methoxyethyl) (2′-MOE)-modified antisense oligonucleotide can also silence the TTR gene to treat the hereditary transthyretin amyloidosis [126]. Recently, a newly designed in vivo CRISPR-Cas9-based gene-editing therapeutic agent, NTLA-2001 was introduced. It comprises of a lipid nanoparticle that is encapsulating mRNA for Cas9 protein and a guide RNA targeting the TTR gene. From the administration of NTLA-2001 in six patients, only mild adverse events were reported. All the patients showed decreased serum TTR protein concentrations through the targeted knockout of TTR (NCT04601051) [127].

### 4.2. Acute Intermittent Porphyria

Acute intermittent porphyria (AIP) is an autosomal dominant metabolic disorder that is caused by the deficiency of porphobilinogen deaminase (PBGD), an enzyme that is involved in the synthesis of heme [128]. The δ-aminolevulinate synthase 1 (ALAS1) is the first enzyme and rate-limiting step in the synthesis of heme. PBGD is the third enzyme in the cascade of heme synthesis and it is encoded by the human HMBS gene. Mutations in the HMBS gene lead to the reduction of PBGD activity and reduce the level of heme. This affects the normal feedback of the hepatic ALAS1 expression. The down-regulation of the PBGD activity resulted in the up-regulation of the ALAS1, causing a marked overproduction and accumulation of the precursors δ-aminolevulinic acid (ALA) and porphobilinogen (PBG) [129]. Clinically, AIP is characterized by repetitive acute neurovisceral attacks with a high level of neurotoxic porphyrin precursors. Patients with AIP may develop severe fatigue, abdominal pain, loss of appetite, nausea, vomiting, and constipation. A few constitutional symptoms such as sleep disturbance, anxiety, depression, and mental confusion can be observed. Severe neurological complications occasionally take place and death may be caused by respiratory or bulbar muscle paralysis [130].

The infusion of hemin and loading of carbohydrates are the frequently used treatments for AIP. The supply of hemin can restore the heme pool in the liver and also suppress the ALAS1 expression [131]. About 5% of AIP patients suffer recurrent attacks which may persist for many years [132] and usually requires hospitalization and strong analgesic treatment. The prophylactic use of hemin increases among the AIP patients with repeated chronic symptoms. However, repeated infusion of hemin can generate thromboembolic events and increase the serum level of ferritin which may cause iron overload in the liver [133].

An alternative treatment strategy utilizes a short interfering RNA to reduce the level of *ALAS1* transcript [134]. The application of lipid nanoparticle-formulated ALAS1-targeting siRNA (Givosiran) has recently been proven to be safe and effective in relieving symptoms as well as in decreasing the accumulation of porphyrin precursors among AIP patients [135,136,137].

### 4.3. Spinal Muscular Atrophy

As mentioned earlier, utilizing ASO can specifically decrease the level of the targeted protein. Another class of oligonucleotide therapeutics are single-stranded splice-switching oligonucleotides (SSOs) which target the pre-mRNA of interest in the nucleus and alter its splicing process to produce a functional protein [138]. In 2016, the FDA approved two SSOs to treat monogenic neurological diseases; Spinraza^®^ (2′-O-methoxyethylphosphorothioate) (Biogen, 2016) and Eteplirsen^®^ (phosphorodiamidate morpholino oligonucleotide, PMO) [139] for the treatment of spinal muscular atrophy and Duchenne muscular dystrophy, respectively. Several SSOs have been developed recently, among which morpholino oligonucleotide (PMO) is the most commonly used. Järver et al. developed lipid nanoparticles that can be efficiently exploited for cellular transfection of the charge-neutral oligonucleotides such as the PMO based splice-switching oligos [140].

Spinal muscular atrophy (SMA) is an autosomal recessive neuromuscular disorder that is also the leading inherited monogenetic cause of death in infants. The motor neuron degeneration in SMA causes marked weakness and atrophy in the limbs and bulbar muscles [141]. A deletion or point mutation in the survival motor neuron 1 (SMN1) gene is responsible for SMA. The mutation leads to the reduction of SMN protein level with subsequent degeneration of motor neurons in the anterior horn of the spinal cord and brainstem [142]. The SMN2 gene is nearly identical to SMN1 and can serve as a backup for the SMN1 gene [143]. Position c.804 is critical for the correct splicing of the SMN pre-mRNAs. The c.804C in the SMN1 pre-mRNA bestows the correct splicing signal to produce full-length functional SMN protein, whereas c.804T renders aberrant splicing that is associated with the exon 7. Some SMA patients carry the mutation(s) in the SMN2 gene and c.804T mutation resulted in the deletion of exon 7 in nearly 90% of the SMN2 transcripts (Δ7SMN2) [143]. Using a steric-blocking PMO to bind the pre-mRNA of the mutated SMN2 gene can successfully block the aberrant splicing event, produce full-length functional SMN2 [144], and compensate for the deficiency of SMN1. Thus, these steric-blocking PMOs provide a significant therapeutic benefit for SMA patients. The US FDA approved Nusinersen (Spinraza^®^) in 2016 for the treatment of all types of SMA [145].

In addition to the SSO to block the aberrant splicing, gene-replacement therapy (Onasemnogene abeparvovec) that is based on adeno-associated virus serotype 9 (AAV9) to provide the full-length SMN1 protein has been designed. Increased SMN protein expression following the treatment has successfully prevented the death of motor neurons with the improvement of neuromuscular functions in the affected children. The phase 3 trial provides promising outcomes and Onasemnogene abeparvovec (Zolgensma^®^) was approved by the FDA in 2019 [146,147]. Recently, a matching-adjusted indirect comparison study between Onasemnogene abeparvovec and Nusinersen found that, in terms of event-free survival, overall survival, and motor milestone achievement, Onasemnogene abeparvovec provides a continuous benefit compared to Nusinersen in 24 months follow-up [148].

### 4.4. Duchenne Muscular Dystrophy

Duchenne muscular dystrophy (DMD) is one of the most common inherited muscular disorders. DMD is caused by the mutations in a large dystrophin coding gene (DMD), 3685 amino acids long from 79 exons spanning more than 2Mb in the genome [149]. Most of the mutations in the DMD gene result in the production of a non-functional dystrophin protein. Progressive and massive muscle fiber degeneration are the common clinical presentations that are associated with DMD. Mutations involving a part of the central rod domain of the dystrophin protein can still produce a partially functional protein that leads to a milder condition of the DMD, Becker muscular dystrophy (BMD) [150]. Therefore, it is conceivable to restore part of the dystrophin protein to reduce the severity of DMD. This presumption provides the basis for a DMD treatment through the modulation of the pre-mRNA splicing [151]. Deletion mutations in the exons of the DMD gene frequently cause a frameshift that leads to the incorporation of a premature stop codon, which promotes mRNA degradation via the nonsense-mediated decay pathway [152]. The majority of pathogenic variants in the DMD gene are clustered between the exons 45 and 55. The DMD clinical phenotype highly correlates with the DMD genotype [153,154,155]. For example, patients with mutations in exon 51 or 53 have a quick decline in walking ability, lose their ambulation at an early age, and have poor pulmonary outcomes as compared with the patients who harbor the mutations in exon 44 [153,154,156]. Based on the observation, to convert an out-of-frame transcript into an in-frame transcript utilizing the antisense-mediated exon skipping treatment strategy may ameliorate the disease severity of DMD [157]. There were two clinical trials using drugs, Drisapersen (from GSK) and Eteplirsen (Sarepta) that were carried out for patients who are amenable by exon 51 skipping which represents about 13% of all DMD patients. Drisapersen is based on 2′OMePS chemistry whereas Eteplirsen is using PMO chemistry. The report from PROMOVI trial, a phase 3, multi-center, open-label study for Eteplirsen suggests that there is a clinically notable attenuation of decline on the six minute walk test and significant reduction of percent predicted forced vital capacity annual decline in the DMD patients [158]. Even though the phase 3 trial for Drisapersen did not meet its primary endpoint, the phase 2 trial did show beneficial outcomes for the DMD patients (PMC:6093847). Recently, another SSO, Golodirsen that is based on PMO chemistry that was designed for DMD patients amenable for exon 53 skipping has been approved as an effective drug [159,160].

### 4.5. Huntington’s Disease and Other Neurodegenerative Disorders That Are Caused by Microsatellite Repeat Expansion

As aforementioned, microsatellite repeat expansion (MRE) in both coding and non-coding regions of the genome has been associated with many neurological disorders, such as Huntington’s disease (HD), spinocerebellar ataxia, Friedreich ataxia, myotonic dystrophy, frontotemporal dementia (FTD), and amyotrophic lateral sclerosis (ALS) [161]. The hallmark for these MRE-associated disorders is the disturbances in the RNA metabolism resulting from the expansion of the repeat sequence which affects the proper translation and protein degradation [162]. Uncovering how MRE results in neuropathologies may shed light on the strategies to develop versatile therapeutics [162]. It is conceivable that the design of these therapies may be readily adaptable to a variety of MRE disorders since the post-transcriptional pathologies of many MRE disorders are similar.

HD is an autosomal dominant neurodegenerative disorder resulting from a mutation in the human huntingtin (HTT) gene. The clinical features include dementia, depression, schizophrenia, hyperkinetic movements such as chorea and athetosis, oculomotor apraxia, bipolar disorders, and sometimes suicidal events [163]. There is no cure for HD but several FDA-approved drugs such as tetrabenazine (Xenazine), deutetrabenazine (Austedo), haloperidol, risperidone (Risperidal), olanzapine (Zyprexa), quetiapine (Seroquel), amantadine, levetiracetam (Keppra), and clonazepam (https://www.ninds.nih.gov/Disorders/All-Disorders/Huntingtons-Disease-Information-Page, accessed on 18 November 2019) have been used to relieve the symptoms. The expansion of the CAG trinucleotide repeats in the HTT gene leads to an extended polyglutamine in HTT which promotes protein aggregation—the pathogenic culprit of HD.

Recently, an RNA-based target therapy has been developed utilizing ASOs which binds to the HTT transcript followed by the activation of an RNase H-mediated HTT mRNA degradation [164]. The copies of wild-type and mutated HTT transcripts are reduced after the application of non-selective ASOs, whereas selective ASOs can target only mutant allele HTT transcripts. The main advantage of selective ASO is to preserve the wild-type transcript. However, to be allele-specific, the selective ASOs need to be designed to target single nucleotide polymorphisms (SNPs) that are only present in the mutated allele [165]. Since there are different SNPs in patients from different ethnic or geographical origins, selective ASO-based therapy is only effective in those who carry particular SNPs [166]. As mentioned, the application of non-selective ASO results in the downregulation of both the wild-type and mutated HTT transcripts. Studies from both rodent and non-human primates demonstrated that a partial reduction of wild-type HTT would not cause any changes in motor function or alterations in the brain [167]. However, whether the reduction of wild-type HTT after long-term use of non-selective ASO has any adverse consequences, remains elusive. There are two ASO drugs for HD that are currently in clinical trials. A non-selective ASO for the HTT transcript, Ionis-HTTrx, Tominersen (NCT02519036) has completed phases 1 and 2a [168] with no serious adverse events reported and there was a significant dose-dependent reduction of mutated HTT transcript in CSF [168]. Tominersen has now entered a large multi-center international efficacy trial. Another ASO involves the use of two selective ASOs which specifically target the most commonly occurring SNPs in mutated HTT transcripts—WVE-120101 (PRECISION-HD1) and WVE-120102 (PRECISION-HD2) (NCT03225833, NCT03225846). In the trial of PRECISION-HD2, the results from 88 participants revealed no statistically significant changes in mHTT transcript level versus the placebo after single or multiple doses of WVE-120102 and there was no evidence of a dose response across the ranges that were tested. PRECISION-HD1 trials are still ongoing (https://www.globenewswire.com/news-release/2021/03/29/2201081/0/en/Wave-Life-Sciences-Provides-Update-on-Phase-1b-2a-PRECISION-HD-Trials.html, accessed on 10 November 2021). In addition to ASO, another strategy is using microRNA to recruit HTT mRNA to the RNA-induced silencing complex (RISC) which leads to the degradation of the transcripts. In animal studies, AAV9 viral particles carrying miRNA have been designed and delivered via direct intrastriatal administration resulting in a therapeutic benefit [169,170,171] The AAV9-miRNA approach is feasible for both selective and non-selective approaches; nevertheless, the administration is limited only by direct intrastriatal injection. An miRNA directly against HTT, AMT-130 (rAAV5-miHTT) is in an ongoing phase 1/2 clinical trial (NCT04120493).

### 4.6. Lysosomal Storage Diseases

Lysosomal storage disorders (LSD) are a group of rare diseases and more than two thirds of LSD patients present CNS involvement. The defect of lysosomal proteins in LSD results in an accumulation of nonmetabolized pathogenic products within neurons or other cells. For example, Fabry disease is caused by the accumulation of globotriaosylceramide in cells due to alpha-galactosidase A (GLA) deficiency. The accumulation of glucosylceramide due to the lack of beta glucocerebrosidase (GBA) is caused the Gaucher disease. The LSD-induced neurodegeneration may start as early as the first decade of life, and, once triggered, the progression is unrelenting [172]. A few common CNS manifestations for LSD include delayed development, spasticity in movement, hypotonia, encephalopathy, seizure, and cherry red spots in macula [173].

One of the approved therapies for LSD is an enzyme replacement therapy (ERT), by providing a recombinant protein or an enzyme to replace the defective one. Some positive results for ERT in Gaucher, Fabry, and Pompe diseases have been reported [174,175,176,177]. However, since the recombinant enzymes cannot cross the BBB, ERT was limited in patients without CNS pathologies. Moreover, the body’s reactions toward the infused protein resulted in a low half-life with poor availability which attenuate the therapeutic efficacy. To address the enzymatic defects and the downstream consequences, a few alternatives are under development. For example, substrate reduction therapy (SRT) by blocking the synthesis of the upstream substrate subsequently reduces the accumulation of pathogenic products. SRT can be achieved by small interfering RNAs (siRNA) targeting the upstream genes, such as the glucosylceramide synthase (GCS) to reduce the synthesis of glucosylceramide to treat Gaucher disease (PMID:16959503) [178].

Different synthetic nanoparticles, from electrolytic complexes to liposomes and aggresomes have been tested to deliver small RNAs and mRNAs within the last decade [179]. Extracellular vesicles (EVs) have also been used to effectively deliver functional GLA to cells in Fabry disease models [180]. Another class of special liposomes called stable nucleic acid lipid particles (SNALPs) has been found as a promising platform for delivering small RNA molecules [181]. SNALPs are characterized by the high efficiency of cargo encapsulation and can be engineered to target specific receptors. With preliminary success, the SNALPs and other nanoparticles such as EVs showed the potential of delivering RNA therapeutics to CNS in patients with LSDs.

## 5. Limitations and Future Potentials of RNA Therapies

To generate high-purity therapeutic grade RNA molecules with proper chemical modifications is much faster and less expensive than the production of either traditional small molecule drugs or recombinant proteins. In addition, the manufacturing process is independent of the RNA sequence. Compared to DNA-based gene therapy, RNA has a superior safety profile and the regulatory requirements are easy to follow. Moreover, RNA doesn’t integrate into the host genome [182]. There are a few hurdles when translating the RNA-based therapeutics into the clinic, (1) specificity, (2) delivery, and (3) tolerability. For specificity, sometimes the uptake in cells other than the cells from nervous tissues may produce undesired on-target effects. Besides that, the off-target effect usually results from either sequence similarity or an overdose of the RNA drugs. In terms of delivery, a few points need to be addressed: the instability of RNA, the inefficient intracellular delivery, and the efficiency of crossing the BBB to target the nervous tissues. Finally, the tolerability issue is related to the immune reaction that is caused by the activation of pathogen-associated molecular pattern (PAMP) receptors, such as the Toll-like receptors (TLRs). The potential solutions for these hurdles have been described elsewhere [183], but the recent advances have overcome some of those key obstacles such as the evasion of innate immune activation, increase RNA stability, and development of targeted delivery, which facilitate the rapid growth of RNA therapeutics.

Recently, hospital-based RNA therapeutic development has successfully demonstrated the feasibility of generating novel nucleic acid drugs for patients. One of the examples is the investment of National Children’s hospital into cGMP facilities to introduce gene therapies for SMA (onasemnogene abeparvovec) and DMD (golodirsen). Thus, hospital-based programs will promote the development and implementation of personalized medicines. It is reasonable that the hospital generates the cGMP grade RNA therapy and quickly delivers it to the patient, since manufacturing of the drug at such a small scale is rarely justified financially by the big pharma companies. Damase et al. described their effort to quickly develop new constructs in manufactures research and clinical grade RNA therapeutics [184]. Their team members include RNA biologists, bioinformaticians, and nanomedicine experts. The program offers a single-entry point with consultations to ensure a flawless transition among the different development/manufacturing stages. Thus, hospital-based programs may accelerate the development and utilization of RNA therapeutics by integrating the process between research groups, manufacture groups, and clinicians which shortens the time “from bench to bedside.”

## 6. Conclusions

Advances in molecular diagnosis and the development of RNA-based medications reveal a great potential in precision therapy for neurological disorders. One of the major advantages of RNA-based therapy is its flexibility as it can target both coding and non-coding RNAs. It can also work on previously undruggable targets, for example, the knockdown of TTR expression by ASOs. RNA therapeutics can be designed rapidly. As soon as the sequence is established, the next step is to modify the sequence to minimize the degradation and enhance its binding stability.

For some RNA-based therapies, for example, enzyme replacement therapy, it can have long-lasting effects compared to the traditional small molecule-based drugs where the therapeutic effects are completely governed by its ADME profile (absorption, distribution, metabolism, and excretion). To develop long-lasting RNA-based medications to treat or cure patients with a monogenic disease since birth remains attractive to researchers. Apart from the RNA drugs, the delivery system plays a critical role in the success of RNA-based therapies, especially for treatments that are targeting the brain. The recent development of lipid-based nanocarriers and extracellular vesicles provides a possibility to effectively deliver therapeutics passing through the blood brain barrier. In summary, we are witnessing the development of RNA therapies to treat monogenic diseases. Further refinement to optimize the design of therapeutic RNA molecules and the delivery system may shed light on how to treat previously untreatable hereditary neurological diseases.

## Figures and Tables

**Table 1 biomedicines-10-00158-t001:** The potential strengths and limitations for lipid-based nanocarriers in RNA therapy.

Nanocarrier Type	Strengths	Limitations
Lipid nanoparticles	Physically stable Controlled release (possible) Easily upscalable Can load two different drugs, one on the surface and the other in the core	Solid particles may have difficulties encapsulating molecules like RNA Needs further evidence for parental use
Liposomes	Ease in modification and preparation RNA transfection and delivery: many experiences Has been used by parental administration	Expensive Some RNA may not be encapsulated in the lipoplexes Reduced stability with the risk of immunogenicity
Nanoemulsions	Ease in preparation Potentials to load multiple therapeutic agents	The liquid formulation may be less stable Large droplet size

**Table 2 biomedicines-10-00158-t002:** Clinically approved RNA drugs for single-gene neurological disorders.

Drug (Approved Year)	Target Disease	Target Molecule	Mode of Action	Category of RNA Therapy
Nusinersen (2016)	Spinal muscular atrophy	SMN2 mRNA	Modifying alternative splicing of the SMN2 mRNA to increase the SMN protein level	Single-strand antisense
Eteplirsen (2016)	Duchene muscular dystrophy	Dystrophy mRNA	Induce the exon 51 skipping during the splicing process to produce a functional dystrophin protein	Single-strand antisense
Inotersen (2018)	Hereditary transthyretin amyloidosis	Transthyretin mRNA	Complementary binding to the Transthyretin mRNA to induce RNase H-based degradation	Single-strand antisense
Golodirsen (2019)	Duchene muscular dystrophy	Dystrophy mRNA	Inducing the skipping of exon 53 during the splicing process to produce a functional dystrophin protein	Single-strand antisense
Patisiran (2018)	Hereditary transthyretin amyloidosis	Transthyretin mRNA	RNA interference to inhibit the production of transthyretin protein	Double-strand small interfering
Givosiran (2019)	Acute hepatic porphyria	ALAS1 mRNA	RNA interference to suppress the hepatic production of ALAS1 enzyme	Double-strand small interfering
Pegaptanib (2004)	Age-related macular degeneration	VEGF protein	Specific binding to the 165 isoforms of VEGF to inhibit its function	RNA aptamer

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
