# Peer review of "Recent Advances in RNA Therapy and Its Carriers to Treat the Single-Gene Neurological Disorders"

_biomedicines, 2022, doi:10.3390/biomedicines10010158_

Round 1

Reviewer 1 Report

In this review article, the authors provide an overview of current RNA therapies for single-gene neurological disorders treatment and discuss its advantages, limitations, and clinical application. However, the article is not well organized. Please see below for comments:

  1. The logical flow needs to be strengthened. The change between sections is too abrupt and transition sentences or paragraphs are needed.
  2. I highly suggest adding figures or tables to summarize approaches of RNA therapy and to interpret the mechanism of different RNA therapies.
  3. The clinical applications of RNA therapies were only briefly mentioned. Please further discuss the clinical performance of those applications and is there any RNA therapy approved for commercial use?
  4. In section 4, the title is, “Using nanocarriers to treat single-gene neurological disorders.” Why only focus on the utilization of the nanocarrier? What about other delivery strategies for treatment? From what the authors presented; it’s not convincing to believe that only nanocarrier steady is worth discussion.
  5. Please further discuss the risks and barriers of RNA therapy. For example, how to monitor or control the expression after delivery of the RNA? How about the potential long-term complications?
  6. Please further discuss the future potential of RNA therapy applications.

Author Response

Response to Reviewer 1 Comments

We thank the reviewer for the very helpful suggestions on the manuscript. We have carefully revised the manuscript according to the reviewer’s comments.

  1. The logical flow needs to be strengthened. The change between sections is too abrupt and transition sentences or paragraphs are needed.

Response:

We have added a few sentences in the revised manuscript to strengthen the logic flow and smooth the transition.

  1. I highly suggest adding figures or tables to summarize approaches of RNA therapy and to interpret the mechanism of different RNA therapies.

Response:

We thank the reviewer’s helpful comments and have added two tables in the revised manuscript. Table 1 summarized the potential strengths and limitations of lipid-based nanocarriers in section 3.2. Table 2 listed the RNA drugs which have been approved by the FDA. In Table 2, we also included targeted disease, targeted molecule, as well as the mode of action. Table 2 was inserted in section 4. 

  1. The clinical applications of RNA therapies were only briefly mentioned. Please further discuss the clinical performance of those applications and is there any RNA therapy approved for commercial use?

RESPONSE:

We appreciated the reviewer’s suggestion. In preparation for the manuscript, we focused on the treatment of single-gene neurological disorders. RNA therapies for other systemic diseases or medical problems such as vaccination are beyond the scope of this review. In table 2 and the following sections, we delineated the clinical applications of RNA therapies to treat single-gene neurological disorders. Among the approved RNA drugs, they belong to different categories; some are single-strand antisense RNA, others are double-strand small interference RNA. For example, Inotersen suppresses the level of aggregated protein, TTR, whereas Givosiran, RNA interference, inhibits the enzyme ALAS1, which is the first enzyme for the synthesis of heme. The details of their mode of action have been described in the individual sections.

  1. In section 4, the title is, “Using nanocarriers to treat single-gene neurological disorders.” Why only focus on the utilization of the nanocarrier? What about other delivery strategies for treatment? From what the authors presented; it’s not convincing to believe that only nanocarrier steady is worth discussion.

RESPONSE:

We appreciate the helpful comments from the reviewer. We followed the suggestion and change the title of the section to “RNA therapies to treat single-gene neurological disorders”. In this section, both RNA drugs and their associated nanocarriers have been described. In addition to the clinically approved RNA medications, drugs in clinical trials have also been included.

  1. Please further discuss the risks and barriers of RNA therapy. For example, how to monitor or control the expression after delivery of the RNA? How about the potential long-term complications?

RESPONSE:

Thank you for the comments! A description of the limitations and potential problems for RNA therapy has been added.

  1. Please further discuss the future potential of RNA therapy applications.

RESPONSE:

The clinical applications, especially the hospital-based development programs to generate RNA therapy have been added.

Reviewer 2 Report

The review article comprehensively describes the types of RNA-based therapy approaches, drug delivery techniques aiming treatment of the number of neurological diseases.

Comments

  1. As the article contains profound description of the drug delivery techniques, this issue should be indicated in the Title.
  2. Line 411: “TRR transcript”. All abbreviations should be disclosed.

Author Response

Response to Reviewer 2 Comments

We thank the reviewer for the helpful advice on the manuscript. We have carefully revised the manuscript according to the reviewer’s comments.

  1. As the article contains profound description of the drug delivery techniques, this issue should be indicated in the Title.

Response:

We thank the reviewer`s comments and have updated the title to “Recent advances in RNA therapy and its carriers to treat the single-gene neurological disorders”.

  1. Line 411: “TRR transcript”. All abbreviations should be disclosed.

Response:

We apologize for the typo. It should be “TTR” and we have corrected it. The abbreviation has been mentioned in the previous paragraph.

Round 2

Reviewer 1 Report

The authors have addressed my concerns.